# Estimating Ground Reaction Forces from Two-Dimensional Pose Data: A Biomechanics-Based Comparison of AlphaPose, BlazePose, and OpenPose

**DOI:** 10.3390/s23010078

**Published:** 2022-12-21

**Authors:** Marion Mundt, Zachery Born, Molly Goldacre, Jacqueline Alderson

**Affiliations:** 1UWA Minderoo Tech & Policy Lab, Law School, The University of Western Australia, Crawley, WA 6009, Australia; 2Sports Performance Research Institute New Zealand (SPRINZ), Auckland University of Technology, Auckland 1010, New Zealand

**Keywords:** motion analysis, biomechanics, pose estimation, machine learning, artificial neural networks

## Abstract

The adoption of computer vision pose estimation approaches, used to identify keypoint locations which are intended to reflect the necessary anatomical landmarks relied upon by biomechanists for musculoskeletal modelling, has gained increasing traction in recent years. This uptake has been further accelerated by keypoint use as inputs into machine learning models used to estimate biomechanical parameters such as ground reaction forces (GRFs) in the absence of instrumentation required for direct measurement. This study first aimed to investigate the keypoint detection rate of three open-source pose estimation models (AlphaPose, BlazePose, and OpenPose) across varying movements, camera views, and trial lengths. Second, this study aimed to assess the suitability and interchangeability of keypoints detected by each pose estimation model when used as inputs into machine learning models for the estimation of GRFs. The keypoint detection rate of BlazePose was distinctly lower than that of AlphaPose and OpenPose. All pose estimation models achieved a high keypoint detection rate at the centre of an image frame and a lower detection rate in the true sagittal plane camera field of view, compared with slightly anteriorly or posteriorly located quasi-sagittal plane camera views. The three-dimensional ground reaction force, instantaneous loading rate, and peak force for running could be estimated using the keypoints of all three pose estimation models. However, only AlphaPose and OpenPose keypoints could be used interchangeably with a machine learning model trained to estimate GRFs based on AlphaPose keypoints resulting in a high estimation accuracy when OpenPose keypoints were used as inputs and vice versa. The findings of this study highlight the need for further evaluation of computer vision-based pose estimation models for application in biomechanical human modelling, and the limitations of machine learning-based GRF estimation models that rely on 2D keypoints. This is of particular relevance given that machine learning models informing athlete monitoring guidelines are being developed for application related to athlete well-being.

## 1. Introduction

The holy grail of sports biomechanics is to enable high-fidelity motion analysis outside the laboratory without inhibiting the athlete’s movement or performance. Although there are a plethora of studies analysing motion, there remains a lack of understanding of the biomechanical load an athlete is exposed to while in training and competition environments. In their efforts to optimise performance and prevent injury, this information is of significant interest to the sport practitioner, especially as it relates to the determination and prescription of optimal training loads [1].

The scientific terms *load* and *workload* in sport and exercise science are broad, simplistic, and regularly misused [2,3]. The commonly used term *training load* refers to an athlete’s physiological and biomechanical stress, sometimes also incorporating psychological stress [1]. *Training load* summarises the *external load* that is prescribed to an athlete, while *internal load* can be considered the athlete’s response to the *external load* [4]. It should be noted that when referring to both *internal* and *external load*, the term *load* can be considered misleading from a purists perspective as *external load* is being used to describe parameters such as linear full-body speed or power, while *internal load* refers to heart rate or self-reported measures of perceived exertion, which are not by definition, mechanical *loads* [2]. Despite this, there appears to be general acceptance in the sport and exercise science community that the physiological and biomechanical components of *training load* comprise exercise volume and intensity [3]. The current paper adopts the GRF parameters of instantaneous loading rate (ILR) and maximum force ($F_{max}$) as descriptors of exercise intensity [5,6].

Laboratory-based and on-field technologies are commonly used to measure ground kinetics. Ground embedded force plates have been used in biomechanics for decades and are the current gold-standard method to quantify external forces during human movement. Tri-axial force transducers are mounted in each corner of the force plate, that determine the three-dimensional (3D) force. The four single forces are then used to determine the resulting force in vertical, medio-lateral and anterior–posterior direction. Further, the measurement of the four sensors in the corners allows for the calculation of the centre of pressure and free moment of rotation applied to the force plate [7]. Although considered the gold standard for measuring GRF, force plates are laboratory-bound, and have poor external ecological validity [8]. Portable technologies, such as pressure insoles, can be used in the field, but these technologies are limited to determine the vertical GRF component and have been reported to interfere with an athlete’s free movement [9]. To address these limitations, previous sports biomechanics researchers have trained machine learning models that seek to estimate GRF from motion kinematics collected using 3D retro-reflective optical motion capture [10,11,12,13], inertial sensors [8,14,15,16], and 2D video based pose estimation keypoints, themselves derived from other computer vision-based machine learning models [17,18]. Since traditional 3D optical motion capture is restricted to a laboratory environment and body-affixed inertial sensors pose a safety risk during competition and may also inhibit athlete motion, the focus of this study was to investigate the utility of 2D pose estimation algorithms for use in biomechanics.

From high speed film to 2D video cameras, the process of capturing field-based vision which is then processed by the manual digitising of anatomical landmarks, has underpinned the discipline of sports biomechanics for more than half a century. This digitisation process is labour intensive, time-consuming, and prone to human error [19]. Recent advances in compute power and computer vision techniques such as pose estimation, means the process of digitising 2D videos and extracting joint centre and other anatomical landmark information, can be automated [20]. Consequently, there has been rapid development in open-source frameworks containing pose estimation algorithms and pre-trained pose estimation models to automatically determine human pose from video image sequences.

The most commonly employed pose estimation framework in sports biomechanics is OpenPose [21,22] with successful utilisation in sports biomechanics running [23,24], sprinting [19], rowing [25], cycling [26], and squash [27] research. Another frequently used pose estimation framework in biomechanics and animal tracking is DeepLabCut [28], where users are required to fine-tune a pre-trained pose estimation model by providing the algorithm with a subset of about 200 manually labelled images specific to the desired task and tracking locations [29], e.g., in swimming to detect shoulder, hip, and knee joint centres while the swimmer is in the glide phase as per the work of Papic et al. [30]. The limitation of DeepLabCut’s approach is that a substantial amount of manual digitising labour is required before the model can be employed.

There is no study exploring the between pose-estimation reliability of joint centre locations derived from 2D image frames. This may be attributed to, first, the majority of pose estimation models are trained on datasets that are not intended to be used for biomechanical analysis. These datasets have low accuracy in the manually digitised training images which causes subsequent downstream keypoint detection inaccuracies. Second, the occlusion of limbs and missing depth information in 2D images might result in confusion between the left and right limbs as the pose estimation models try to identify bilateral lower limb keypoints [20,28]. It is clear that further research is required to quantify, understand, and overcome errors in 2D pose estimation.

The aims of this paper are two-fold. First, we will provide an overview of different pose estimation approaches (OpenPose, AlphaPose, BlazePose), the computational efficiency, and challenges we encountered with the different pose estimation frameworks. Second, we will investigate the keypoint detection rate of the three pose estimation models in videos showing different movements and camera views. We will further investigate the interchangeability of keypoints estimated by the different algorithms for the estimation of 3D GRF in running tasks from a true sagittal camera view using an artificial neural network. We focus on the true sagittal camera view because it is used most frequently as a best practice orthogonal-to-plane-of motion camera positioning in biomechanics setups.

## 2. Pose Estimation Algorithms

A variety of algorithms can be used to estimate human pose in 2D videos with the most powerful image processing models relying on convolutional neural network (CNN) architectures for state-of-the-art human pose estimation methods. Early pose estimation algorithms were used to annotate 2D keypoints of a single person in an image frame with models recently extended to detect multiple persons and to determine landmarks in pseudo-3D space (not six degree of freedom) based on single or multiple camera views [20]. Two predominant approaches are used to detect multiple persons in one image—top-down or bottom-up. The top-down approach detects a person in an image first and estimates the different segments of a person second. The bottom-up approach detects the segments of persons first and associates those with individual persons in a second step before finally detecting keypoints [31,32]. Following is an overview of the varying pose estimation algorithmic approaches used in this study.

Bottom-up approaches extract features belonging to segments of the human body from an image using the first few layers of their CNN architecture. In the OpenPose algorithm [31], these features are fed into two parallel branches of convolutional layers. The first branch predicts a set of 18 confidence maps, with each map representing a particular part of the human pose skeleton. The second branch predicts a set of 38 Part Affinity Fields which represent the degree of association between parts. Using the part confidence maps, bipartite graphs are formed between pairs of parts. Weaker links in the bipartite graphs are cut off using the Part Affinity Field values, and keypoints can be assigned to single persons. Multiple calibrated cameras are necessary for the reconstruction of keypoints in 3D. A calibration and reconstruction algorithm is provided in the OpenPose toolbox.

Top-down approaches are usually based on the precision of the person detector, since the pose estimation is conducted on the area where the person is present. Hence, errors in localisation and replication of bounding box predictions will result in erroneous keypoint estimation. The developers of AlphaPose [32] overcame this problem by adding a Symmetric Spatial Transformer Network to extract a high-quality single person region from an inaccurate bounding box. Subsequently, a Single Person Pose Estimator is used in this region to estimate the human pose keypoints, followed by a spatial de-transformer network to remap the estimated keypoints to the original image coordinate system. Finally, a parametric pose non-maximum suppression technique is used to handle the issue of redundant poses. There is no code available to estimate 3D pose, but the use of multiple calibrated cameras can reconstruct 3D pose using a custom algorithm.

BlazePose [33] was developed by Google Research to estimate a single person’s pose, particularly for fitness applications. Pose estimation is undertaken in two steps: first, a detector locates the pose region-of-interest within the image, second, a tracker estimates the keypoints from this region-of-interest. In contrast to other pose estimators, this solution requires an initial pose alignment. Therefore, either the whole person, or at least hip and shoulder keypoints need to be confidently annotated. Keypoints can be estimated in 2D or 3D from a single camera view.

At present, there is no clear guidance on whether one pose estimator is superior over another. OpenPose and AlphaPose can be used for multi-person detection, while BlazePose is restricted to a single person in an image. All algorithms can be used in real-time with BlazePose specifically designed for sport and fitness applications and reported to be optimised for smart phone use.

One major distinction between the three pose estimation models investigated in the current paper is the number and location of automatically identified keypoints (Figure 1). These differences can be attributed to the training dataset used in each model. We used OpenPose and AlphaPose models trained on the COCO 2017 dataset which contains 64,115 images with at least one human with 17 keypoint annotations [34]. BlazePose can only detect a single person in an image and always outputs 33 keypoints. It is trained on a superset of BlazeFace [35], BlazePalm [36], and COCO dataset [34]. The dataset consists of 60K images with a single or few people in the scene in common poses and 25K images with a single person in the scene performing fitness exercises.

## 3. Materials and Methods

### 3.1. Pose Estimation Frameworks

All pose estimation models were run on a Desktop PC with an Intel Core i9 Processor (16 × 3.6 GHz), NVIDIA GeForce RTX 2080 graphics card, and 32 GB RAM running an Ubuntu 20.04 operating system. During processing our dataset (Section 3.3), AlphaPose used 63% of the available 8 GB GPU memory to achieve a frame rate of 16.62 FPS and OpenPose used 92% of the available GPU memory to achieve a frame rate of 12.52 FPS. BlazePose did not use any GPU memory and achieved a frame rate of 8.02 FPS, which resulted in a processing time more than twice as long as AlphaPose.

### 3.2. Video Processing

Based on their documentation, OpenPose and AlphaPose are able to run on videos of any format. All necessary video processing is handled internally which makes the frameworks easy to use. The user only needs to add the paths to the input videos and output results. However, this ease-of-use also hinders quality checks. We found an offset of up to 15 frames for the estimated keypoints when overlaying video and keypoint output outside the pose estimation framework. This problem only occurred when using videos in .MTS file format. The conversion to any standard video file format such as .avi or .mp4 de-interlaces the video and keypoints estimated in those videos did not result in any offset. BlazePose was most difficult to use due to a lack of documentation for setup questions. Further, BlazePose can only handle single image frames, not video data. To run BlazePose, all videos had to be converted to single frames using the OpenCV package in Python. It is worth noting that OpenCV uses a BGR image format while the current standard is an RGB format which makes the transformation not trivial. All keypoints estimated by BlazePose showed an offset of one frame after processing. This was corrected in all further processing steps.

### 3.3. Dataset and Experimental Setup

The full dataset comprised of running, walking, unplanned 45${}^{\circ }$ sidestepping, and cross-over trials (10 trials per movement type), collected from 14 professional and semi-professional female Australian Rules Football Players (23 ± 3.74 years, 62.77 ± 5.41 kg, 168 ± 4 cm). Data were collected at The University of Western Australia and the study was approved by the University’s Human Ethics Committee (approval number: RA/4/1/2593). All participants provided written consent for data collection, analysis and publication. Video data were recorded using three 2D video cameras (Sony HDR-CX700, 25 FPS, 1920 × 1080 px) that were positioned sagittal to the plane of movement: slightly posterior to true sagittal camera location, true sagittal location, and slightly anterior to true sagittal location. The posterior and anterior cameras were panned approximately 30${}^{\circ }$ such that the central location of the camera field of view was the force plate (Advanced Mechanical Technology Inc., Watertown, MA, USA, 2000 Hz, 1200 × 1200 mm). All three 2D camera fields of view were positioned to align with the volume directly above the ground embedded force plate, where the participant stance phase of interest occurs (Figure 2).

Participants approached the force plate at a speed of 4.5–5.5 m s^−1^ for running, sidestepping and cross-over tasks determined by timing gates (SmartSpeed Pro, Fusion Sports, QLD, Australia) positioned 3 m and 0.5 m in front of the force plate in the approach runway. The desired moving direction (left, right, straight) was indicated by an arrow displayed on a projector screen which was triggered as the participant crossed the timing gates 0.5 m immediately prior to force plate contact. Additionally, walking trials at a self-selected speed have been recorded. Due to camera recording errors, data of four participants were excluded from the analysis. The overall number of valid trials per camera view and motion task is displayed in Table 1.

### 3.4. Detection Rate Analysis

The keypoints of all pose estimation models were summarised in single, standardised files for each trial. Stance phases plus 15 lead-in and 10 lead-out frames to include at least one full step before and after the contact of the force plate were extracted from all keypoint files based on manually detected heel strike and toe off events (Figure 3).

Missing keypoints were interpolated using linear interpolation. Note that keypoints missing at the beginning or end of the time window of interest could not be interpolated. The time series was filtered using a 3rd-order Savitzky–Golay filter with a window length of five frames and upsampled to 200 Hz. The time series were upsampled to enable synchronisation with the force data; this step does not influence the quality of the data. Only keypoints of bilateral hip, knee, ankle, and shoulder joint centres were considered for further analysis. These keypoints were translated to a moving coordinate system with an origin at the mid-point of the hip (Figure 4).

The detection rate of the different algorithms was determined by the extracted keypoint data: (1) number of trials containing all keypoints for every frame (after interpolation) for the stance phase including 15 lead-in and 10 lead-out frames and (2) number of trials containing all keypoints for every frame (after interpolation) for the stance phase only. The number of trials per movement and camera view varied (Table 1).

### 3.5. Artificial Neural Network Application—GRF Estimation

The 3D GRF was estimated for running tasks using the true sagittal camera view because it is used most frequently as a best practice camera positioning in biomechanics setups. The stance phase in the GRF data was extracted based on a threshold of 20 N in the vertical force component. Lead-in and lead-out frames were extracted similar to the keypoints accounting for the differing sampling frequency. GRF was downsampled to 200 Hz and small differences in frame numbers between GRF and keypoints were resolved by resampling the GRF.

Only the stance phase of each trial was considered for training an artificial neural network (ANN). The keypoint and GRF data were time-normalised to 101 time frames. A bi-directional long short-term memory (LSTM) neural network was used. The architecture and hyperparameters for all ANNs were the same: an input layer with 101 neurons, two hidden layers with 400 neurons with a dropout rate of 0.5, and an output layer with three neurons. The learning rate was set to 0.003. All networks were trained for 40 epochs and the weights of the epoch with the smallest validation loss were chosen for testing.

Three different test cases were investigated (Figure 5):#1an ANN was trained on keypoints of a single pose estimation model and tested on keypoints estimated by the corresponding pose estimation model;#2an ANN was trained on keypoints of a single pose estimation model and tested on keypoints estimated by the other pose estimation models;#3an ANN was trained on keypoints of all pose estimation models and tested on keypoints of each pose estimation model separately.

All training and testing was undertaken using a leave-one-subject-out (LOSO) cross-validation. This resulted in 40 trained ANNs; the same 30 for test case #1 and #2 and another ten for test case #3. Test case #1 demonstrated which model performs best for the estimation of GRF—which pose estimation model is recommended to use when aiming to estimate GRF from video data without any previous work. Test case #2 showed whether pre-trained models can be used for keypoints estimated by a different pose estimation algorithm. This can be useful if a model trained on keypoints of one algorithm is available but in previous work keypoints have been estimated using a different pose estimation model. Test case #3 identified if it is useful to train a model on different pose estimation keypoints for an overall higher estimation accuracy.

The final dataset contained 97 unique GRF samples and three keypoint samples (true sagittal camera view determined by three pose estimation algorithms) for each GRF sample. This resulted in an input matrix size [n × 101 × 16] and an output matrix of size [n × 101 × 3] with n=97 for test case #1 and #2, and n=291 for test case #3.

### 3.6. Data Analysis

Ground reaction force estimation accuracy was analysed using the correlation coefficient (*r*) and the normalised root-mean-squared error (nRSMSE). GRF time series were further analysed using statistical parametric mapping (SPM, spm1d.org). Paired-sample *t*-tests established the differences between the estimated and ground truth GRF for each test case.

As estimation of exercise intensity parameters—maximum GRF ($F_{max}$) and instantaneous loading rate (ILR)—are of interest to the applied practitioner, we analysed $F_{max}$ and ILR using paired-sample t-tests (α=0.05) and Cohen’s effect size. Effect size was assessed as: <0.20 trivial, 0.20–0.60 small, 0.60–1.20 moderate, 1.20–2.0 large and >2.0 very large [37]. $F_{max}$ was determined as the maximum of the absolute resultant GRF. The ILR was calculated for the linear slope of the vertical GRF after foot strike [38].

## 4. Results

### 4.1. Detection Rate

Figure 6 shows the number of trials where each pose estimation model detected all eight keypoints. Missing keypoints were interpolated if a keypoint was in the first and last relevant frame. Hence, only trials missing keypoints at the beginning or end were considered not valid. OpenPose and AlphaPose showed very similar detection rates with 94.5% and 98.4%, respectively, for trials including stance phase plus 15 lead-in and 10 lead-out frames. BlazePose only detected all eight keypoints in 65.2% of trials. Interestingly, the detection rate of BlazePose for the posterior sagittal view (89.1%) and walking (84.8%) was higher than for the other views and faster motion. All pose estimation models showed the lowest detection rate in the true sagittal view for the faster movements (AlphaPose 90.4–95.9%, BlazePose 25.0–38.8%, OpenPose 68.7–87.8%). AlphaPose and OpenPose detected all keypoints from all camera views for walking. All three pose estimation algorithms showed a very high detection rate (OpenPose 100%, AlphaPose 100%, BlazePose 99.6%) when analysing only the frames of the stance phase.

### 4.2. GRF Estimation

A bi-directional LSTM neural network was trained to estimate the GRF from keypoints for running tasks. Architecture and hyperparameters were the same for all input datasets. The mean correlation coefficient, *r*, ranged from 0.638 to 0.806 and nRMSE from 0.27 to 0.50 (Table 2). This result was influenced by the estimation of the medio-lateral GRF component (−0.074<r<0.456, 0.64<nRMSE<1.14). The anterior–posterior and vertical GRF components showed high accuracy (r>0.937, nRMSE <0.20). The result of one example trial is shown in Figure 7. High accuracy for all keypoint inputs was found for test cases #1 and #3. Test case #2 showed a decreased accuracy when testing models trained on AlphaPose, or OpenPose on BlazePose keypoints, and testing models trained on BlazePose on AlphaPose or OpenPose keypoints. The accuracy did not decrease when testing a model trained on AlphaPose with OpenPose keypoints and vice versa and therefore, AlphaPose and OpenPose model keypoints can be used interchangeably.

SPM analysis revealed no statistically significant differences between estimated GRF and ground truth for any keypoint input for test cases #1 and #3. For test case #2, differences in the anterior–posterior and medio-lateral GRF component were found when AlphaPose and OpenPose keypoints were used to test the model trained on BlazePose. There were no differences in the vertical GRF component. Differences in all components were found when BlazePose keypoints were used to test models trained on OpenPose or AlphaPose keypoints (Figure 8).

The estimated and ground truth ILR did not differ significantly for any keypoint input. Only the estimation trained on AlphaPose or OpenPose and tested on BlazePose keypoints showed differences larger than 5% and effect sizes of 0.56 and 0.62, respectively. The estimation of the maximum force (Table 3) differed significantly when testing a model trained on AlphaPose and OpenPose on BlazePose keypoints (#2B: AlphaPose p=0.008, d=1.62, OpenPose p=0.019, d=1.34). Differences were also found when a model trained on BlazePose keypoints was tested on AlphaPose (#2A: p=0.027, d=1.12) or OpenPose (#2O: p=0.018, d=1.28) keypoints. All other tested conditions did not show differences exceeding 5% and also returned small effect sizes.

## 5. Discussion

This study investigated the detection rate of three different pose estimation models (Alpha-Pose, BlazePose, and OpenPose) for different movement tasks, camera views, and trial lengths. All pose estimation models returned high detection rates during the stance phase across all walking, running, sidestepping, and cross-over movement conditions. A high keypoint detection rate in the longer videos, including the 15 lead in and 10 lead out frames of stance, was only observed in the walking trials. When movement speed increased, the detection rate decreased, especially for BlazePose. This showed that pose can mainly be detected in frames with the person of interest is located the centre of the image frame, and highlights the limitation of pose estimation in application. Practitioners are not interested in the analysis of a single step only but need information about multiple consecutive steps to gain impactful insights into their analyses.

The AlphaPose model returned the highest detection rate overall, closely followed by OpenPose. Considering the setup time and computational cost, we recommend using AlphaPose for future work from a performance versus computational efficiency perspective.

The true sagittal camera view achieved the lowest detection accuracy overall. One possible explanation for this is that pose estimation models are rarely trained on data that include people in a true sagittal view. Another possible reason is that one side of the body is regularly occluded in this camera view [20,28]. This insight is relevant for practitioners, since a true sagittal camera view is used most frequently as a best practice orthogonal-to-plane-of motion camera positioning in biomechanics setups. Given its methodological importance, to overcome the reduced keypoint detection accuracy for this camera view it may be useful to fine-tune existing pose estimation models with images from a true sagittal view to further improve their applicability to biomechanics.

Our second aim was to analyse whether 2D keypoints output by different algorithms can be used interchangeably. To answer this question, we evaluated the efficacy of GRF estimation in running tasks from a true sagittal camera view using a bi-directional LSTM neural network.

Very good estimation accuracy was achieved for the anterior–posterior and vertical GRF component (r>0.937) but a low accuracy for the medio-lateral component (r<0.456). These results are in line with our previous work that compared different ANNs—a dense neural network and an LSTM neural network—for the estimation of GRF during sidestepping tasks using all three camera views as input data. In the current study, we achieved a slightly higher accuracy than the LSTM neural network with just a single camera view (medio-lateral GRF r=0.128, anterior–posterior GRF r=0.936, and vertical GRF r=0.939) [17]. In another study [18], we synthesised five additional camera views surrounding the force plate and analysed the estimation accuracy for GRF in sidestepping. We achieved high estimation accuracy for all three force components (medio-lateral GRF r=0.926, anterior–posterior GRF r=0.957, and vertical GRF r=0.948). Sidestepping results in larger medio-lateral movement and forces than running, which might simplify the estimation. However, a single camera view does not seem to be sufficient to fully cover 3D GRF estimation. Considering that this study used the true sagittal camera view only, depth information is missing. This might explain the low estimation accuracy in the medio-lateral plane. Adding different camera views and a larger sample size will likely improve this estimation.

To analyse the interchangeability of pose estimation models for the estimation task, we considered three test cases:#1demonstrated which model performs best for the estimation of GRF;#2shows whether pre-trained models can be used for keypoints estimated by a different pose estimation algorithm; and#3identifies if it is useful to train a model on keypoints estimated by different pose estimation algorithms for an overall higher estimation accuracy.

Test cases #1 and #3 did not show any difference in the estimated GRF compared with the ground truth value for all three pose estimation keypoints as test sets. This shows that it is not necessary to train a neural network on keypoints estimated by multiple pose estimation models if only one pose estimation model will be used in future work. For test case #2, we found differences between BlazePose and AlphaPose, and BlazePose and OpenPose, but no differences between AlphaPose and OpenPose. This shows that AlphaPose and OpenPose can be used interchangeably; a model trained on AlphaPose can be used to estimate GRF from OpenPose keypoints and vice versa. Both models are trained on the same dataset (COCO 2017 [34]), which might indicate that other pose estimation algorithms trained on the same dataset might also be used interchangeably and suggests that the dataset the pose estimation model is trained on is more relevant for the output than the algorithm used.

ILR and $F_{max}$ are important biomechanical parameters used to define exercise intensity [5,6] with the evaluation of these parameters necessary to apply the current work in practice. Differences in ILR did not exceed 5% and for $F_{max}$ differences were within 2% for test cases #1 and #3. BlazePose, when not trained and tested on its own keypoints, underestimated ILR by approximately 8% and $F_{max}$ by nearly 15%. These differences are non-trivial when considering the repetitive nature of running and given such underestimations accumulate across a training sessions and competition. In our example, the model was trained and tested on a running speed of 4.5–5.5 m s^−1^ which is considered high-velocity running in females. This is a representative speed found in soccer, Australian Rules Football, and rugby. Dependent on the sport, athletes cover distances of 300 to 1300 m at this speed per match [39]. Considering the distance of a single match, an athlete might take 1000 steps at a high-velocity. A difference of 2 N kg^−1^ for each step would result in a cumulative difference of 2000 N kg^−1^, or 126 kN for our average participant, within a single match. This serves to highlight the importance of achieving a reasonable level of accuracy of such a tool on a single step basis.

The validity of the machine learning model proposed in this study is limited to the population and running speed it was trained on: the dataset comprised only female athletes who ran at a prescribed speed (4.5–5.5 m s^−1^). Future work should validate how the findings translate to a different population and different running speeds. We did not analyse whether the combination of keypoints estimated by two pose estimation models might result in a higher GRF estimation accuracy. Since AlphaPose and OpenPose can be used interchangeably while BlazePose cannot, the incorporation of BlazePose keypoints to the training dataset might have resulted in lower accuracy. Further, future research could investigate how fine-tuning a model that was trained on one set of pose estimation keypoints, using another set of model keypoints, influences the overall estimation accuracy.

## 6. Conclusions

This study aimed to investigate the detection rate of three commonly used pose estimation models across multiple movement types and camera fields of view. A critical finding was that for all pose estimation model keypoints a high detection rate was observed when an individual was located centrally in the image frame, with a lower detection rate observed when the individuals were toward the edges of the image frame. This trend was observed distinctly when using BlazePose. Further, the true sagittal camera view location returned the lowest detection rate compared to the slightly anteriorly or posteriorly located sagittal views. It needs to be investigated whether fine-tuning existing models with more data from a true sagittal camera view can improve performance when using this camera view. If the poor performance is due to occlusion of body parts and cannot be overcome by fine-tuning models with different training data, cameras will need to be set up non-orthogonal to the plane of movement for pose estimation applications—contrary to best practice video capture in biomechanics.

The second aim was to analyse the interchangeability of keypoints determined by different pose estimation models to estimate GRF. All models showed a high estimation accuracy for the GRF when trained and tested on 2D keypoints estimated by the same pose estimation model. AlphaPose and OpenPose keypoints could be used interchangeably—a GRF estimation model trained on AlphaPose keypoints achieved a high estimation accuracy when tested on OpenPose keypoints and vice versa—while the use of BlazePose keypoints for training or testing resulted in an underestimation of GRF. The difference in $F_{max}$ of nearly 15% results in an underestimated biomechanical load of 126 kN for a player weighing 63 kg in a single match. We would recommend AlphaPose for future research, based on our experience regarding ease-of-use, computational efficiency, and detection rate.

The use of machine learning tools that utilise 2D video data inputs is on the rise in the discipline of sports biomechanics and clinical biomechanics. A clear statement of the limitations of a machine learning model is of distinct importance given that they can be used to infer an athlete’s *training load* from broadcast footage with non-valid estimations leading to potentially harmful downstream decision making [40]. Furthermore, the use of remotely obtained video footage to estimate personal information such as load imposes a risk to the privacy and autonomy of an athlete [18,41]. In a clinical context, further validation is necessary with a population showing pathological pose and movement. The transferability of results from this study with a population with healthy pose is most likely limited.

## Figures and Tables

**Figure 1 sensors-23-00078-f001:**
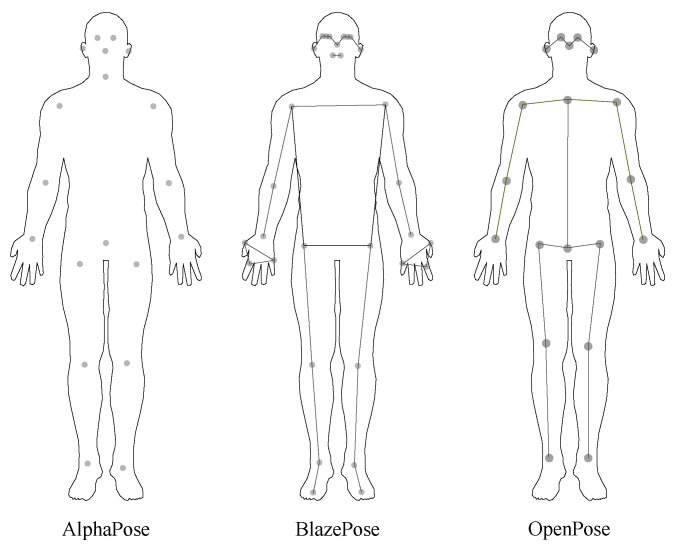
Keypoints determined by the three comparison pose estimation models. AlphaPose and OpenPose detect the same keypoints, as they are trained on the same dataset. BlazePose uses its own dataset for training and detects different keypoints.

**Figure 2 sensors-23-00078-f002:**
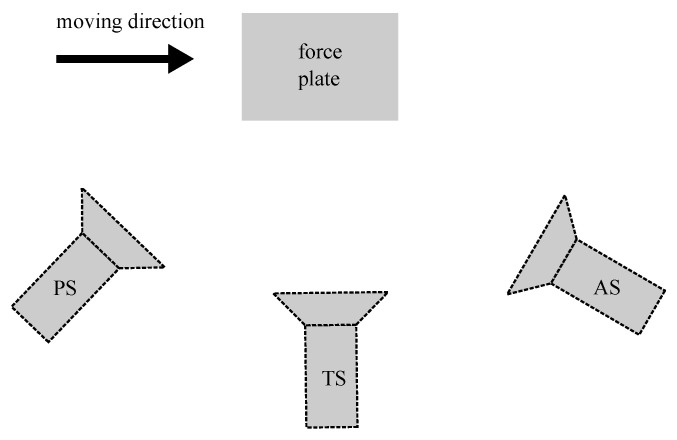
Overview of the experimental setup: three video cameras are positioned sagittal to the plane of motion of the individual, slightly posterior (PS), true sagittal (TS) and slightly anterior (AS).

**Figure 3 sensors-23-00078-f003:**
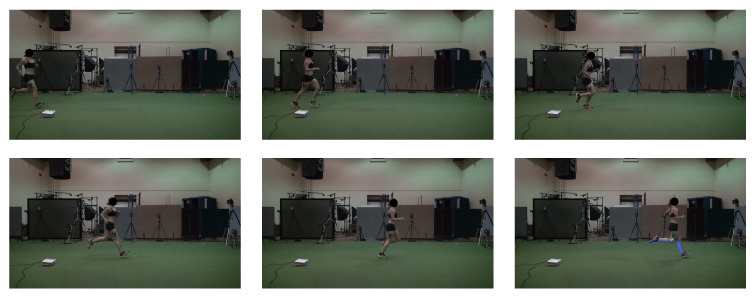
Example of frames including 15 lead-in and 10 lead-out frames.

**Figure 4 sensors-23-00078-f004:**
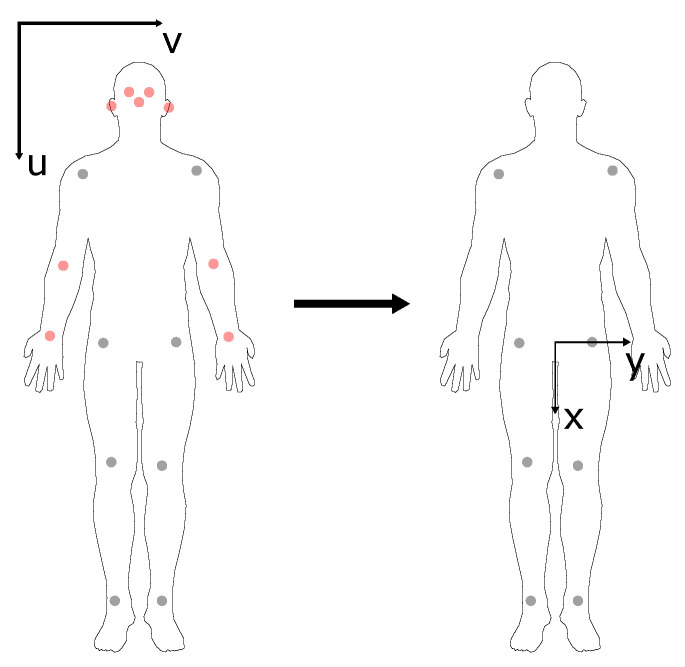
Keypoints estimated by all pose estimation models (**left**). Keypoints displayed in red were excluded from the analysis due to a low detection rate or because they are irrelevant for GRF estimation. The eight keypoints displayed in grey were translated to a moving coordinate system in the mid-point of the hip keypoints (**right**).

**Figure 5 sensors-23-00078-f005:**
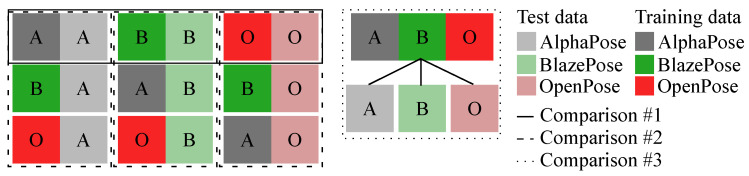
Three different test cases were investigated training and testing ANNs on keypoints estimated by different pose estimation algorithms.

**Figure 6 sensors-23-00078-f006:**
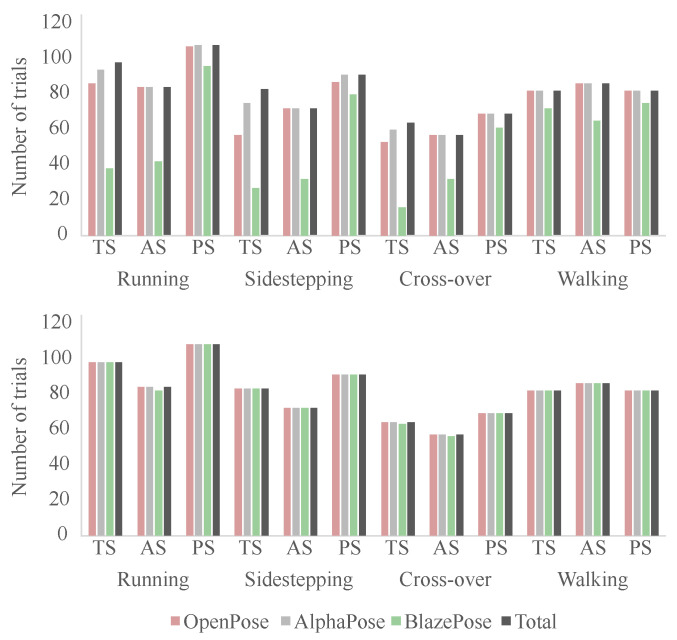
Number of trials in which each pose estimation model detected all keypoints successfully for trials including stance phase and 15 lead-in and 10 lead-out frames (**top**) and just stance phase (**bottom**).

**Figure 7 sensors-23-00078-f007:**
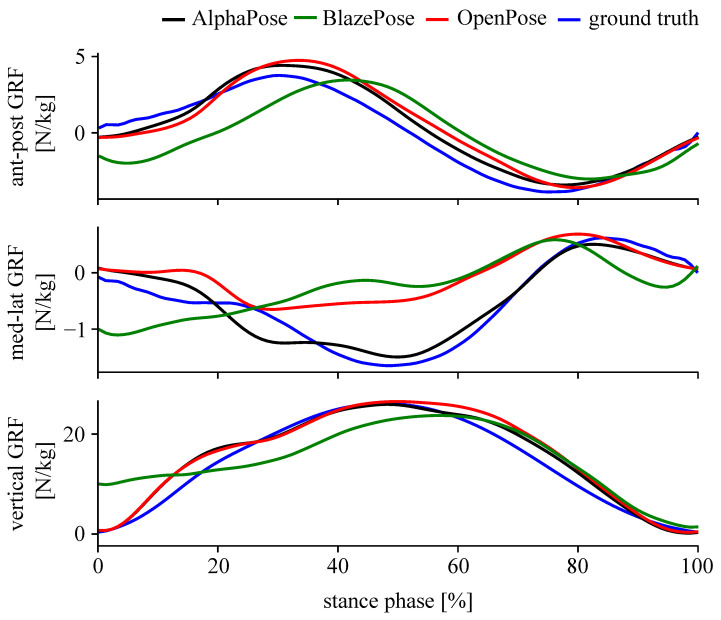
Representative GRF estimation example for a single trial (Test case #3).

**Figure 8 sensors-23-00078-f008:**
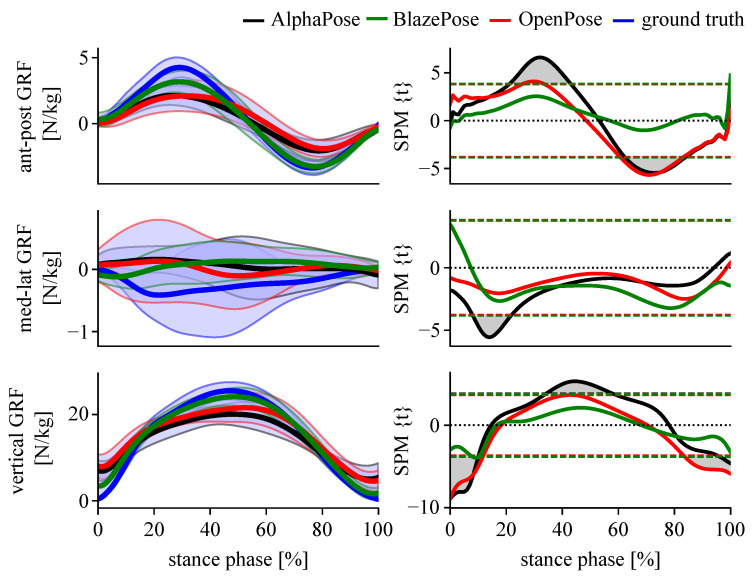
SPM results of BlazePose keypoints tested on AlphaPose, OpenPose and BlazePose keypoints separately (#2) for running tasks.

**Table 1 sensors-23-00078-t001:** Overview of the dataset containing four motion tasks and three camera views (true sagittal TS, anterior sagittal AS, and posterior sagittal PS).

	Running			Sidestepping			Cross-Over			Walking		
**Camera view**	TS	AS	PS	TS	AS	PS	TS	AS	PS	TS	AS	PS
**Trial count**	98	84	108	83	72	91	64	57	69	82	86	82

**Table 2 sensors-23-00078-t002:** Mean correlation coefficient *r* and nRMSE of GRF estimation.

	*r*			nRMSE		
	AlphaPose	BlazePose	OpenPose	AlphaPose	BlazePose	OpenPose
**#1**	0.742	0.698	0.806	0.36	0.30	0.44
**#2A**		0.645	0.759		0.35	0.46
**#2B**	0.638		0.604	0.42		0.50
**#2O**	0.723	0.662		0.36	0.42	
**#3**	0.763	0.681	0.757	0.34	0.27	0.36

**Table 3 sensors-23-00078-t003:** Comparison of $F_{max}$ for the different keypoint inputs. Significant differences are highlighted with a ${}^{*}$.

	AlphaPose			BlazePose			OpenPose		
	$F_{\max}$	$\Delta_{\mathrm{abs}}$	$\Delta_{\mathrm{percent}}$	$F_{\max}$	$\Delta_{\mathrm{abs}}$	$\Delta_{\mathrm{percent}}$	$F_{\max}$	$\Delta_{\mathrm{abs}}$	$\Delta_{\mathrm{percent}}$
	[$\frac{N}{\mathrm{kg}}$]	[$\frac{N}{\mathrm{kg}}$]	[%]	[$\frac{N}{\mathrm{kg}}$]	[$\frac{N}{\mathrm{kg}}$]	[%]	[$\frac{N}{\mathrm{kg}}$]	[$\frac{N}{\mathrm{kg}}$]	[%]
**#1**	24.53	0.46	−1.87	24.43	0.56	−2.25	24.52	0.47	−1.90
**#2A**				22.82 ${}^{*}$	2.16	−9.05	24.97	0.02	−0.07
**#2B**	21.73 ${}^{*}$	3.26	−13.96				22.07 ${}^{*}$	2.92	−12.41
**#2O**	24.12	0.86	−3.52	22.43 ${}^{*}$	2.55	−10.77			
**#3**	25.00	−0.02	0.06	25.15	−0.16	0.65	24.95	0.04	−0.15

## Data Availability

The data presented in this study are available on request from the corresponding author. The data are not publicly available due to source data participant consent limitations.

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
