# Peer review of "Estimating Ground Reaction Forces from Two-Dimensional Pose Data: A Biomechanics-Based Comparison of AlphaPose, BlazePose, and OpenPose"

_sensors, 2022, doi:10.3390/s23010078_

Round 1

Reviewer 1 Report

Topic of the manuscript is very timely as 2d video camera-based pose detection algorithms are increasingly utilized in various motion capture applications.

Introduction is extensive, covering several aspect of the topic.

Methodological Video camera framerate of 25 Hz used in the study is low compared to the commonly used (30, 50, 100 or 120 Hz) frames widely currently, but this does not notably lower the value of the study

Conclusion that the true sagittal view is not best camera placement for pose and joint center detection, is important. The conclusion and the discussion concerning it are of value for the future development of the method and for carrying out measurements.

Details:

In the Figure 5 scale for detection rate is 0...120, and columns of Running, PS look like they exceed 100.  This is confusing layout of the figure. Please clarify what is the value of those largest detection rates, and if it is over 100, explain why.

I'd like to see more clear description, at least an example, of the lead-in and lead-out frames, which sub-phases of the walk cycle are included in the time series when the lead-in and -out frames have been included 

Statistical parameter mapping has bee used in analysis of interchangeably of the three algorithms. It is suitable and increasingly used statistical method for biomechanical time series data.

Author Response

We thank the reviewer for their time and kind comments related to our work, in particular our presentation of figures and schematics. For ease of reference, we have specifically replied to each reviewer comment below, highlighting any undertaken changes in the revised version of the manuscript.

Topic of the manuscript is very timely as 2d video camera-based pose detection algorithms are increasingly utilized in various motion capture applications.

Introduction is extensive, covering several aspects of the topic.

Methodological Video camera framerate of 25 Hz used in the study is low compared to the commonly used (30, 50, 100 or 120 Hz) frames widely currently, but this does not notably lower the value of the study.

Conclusion that the true sagittal view is not best camera placement for pose and joint centre detection, is important. The conclusion and the discussion concerning it are of value for the future development of the method and for carrying out measurements.

Details:

In the Figure 5 scale for detection rate is 0...120, and columns of Running, PS look like they exceed 100.  This is confusing layout of the figure. Please clarify what is the value of those largest detection rates, and if it is over 100, explain why.

Figure 6 in the revised version of the manuscript displays the number of trials on the y-axis, which is exceeding 100. The figure caption was misleading and has been amended. The figure has been updated and additional information has been added in the Methods (Section 3.4).

I'd like to see more clear description, at least an example, of the lead-in and lead-out frames, which sub-phases of the walk cycle are included in the time series when the lead-in and -out frames have been included.

An example has been added (Figure 3).

Statistical parameter mapping has been used in analysis of interchangeably of the three algorithms. It is suitable and increasingly used statistical method for biomechanical time series data.

Reviewer 2 Report

The authors claim to estimate GRF, and not to measure them – under this perspective I read the manuscript. But we have to keep in mind, that estimation is just a measurement with lower demands on what concerns accuracy. And I do not find strainable remarks on the accuracy, nor what concerns the demands ex ante neither for the results ex post the experiments.

For readers from outside biomechanics (the journal is Sensors), some basics on the state of the art of measurement of GRF would be senseful.

From perspective of this state of the art, an effective sampling rate of 8.02 FPS to 16.62 FPS is even to low for analyses of walking, the more for running. This is due to the limitation of sampling frequency of the low budget cameras (I suppose 25 FPS instead of 25 Hz, and fear even those are interlaced), and last this is due to and justified by the limited speed of computers used. Not to talk about the limited spatial resolution due to limited number of pixels. I feel the whole story to be very interesting and to be worth to be told to the public soon, scientific progress is driven by studies like these, but I have some minimum demands:

1.       Discussion on measurement errors of the devices, to have quantitative references for the feasible, and possible perspectives for improvements by technology versus methodological limits. For the ATM this is simple, but the problems with the cameras above are complicated by marker displacement on the skin relative to anatomical landmarks.
Well known in biomechanics, but for the special experiment they have to be quantified. Literature does exist.

2.       Fig. 6 is the only figure which gives „time continuous data“. Due to lacking information at the axes and in the legend, I guess the figure shows data from running trials, and the numbers on the horizontal axes represent time in % of cycle. To give the vertical forces in negative values is just unusal (no discussion here on the equivalence of actio to reactio). The information on ISB or Kistler coordinate system is lacking. This has to be improved to refer to existing mental models on GRF.
And due to the low sampling frequencies, data on walking would be much more interesting than on running what concerns the accurateness – two humps with defined relative heights (the initial spikes and the effects of „wobble masses“ in kinematic data are hidden by compliance).
And we need at least one figure  for an individual walking and running – if Robin Hood in average hits the center, his arrows not necessarily hit the disc.

No reason is given for the upsampling of motion data to 200 Hz – I guess it is a technical means to synchronize and compare kinematic and force data. But you should explicitely annotate, that by interpolation of kinematic data their quality is not increased.

From cryptic fig. 6 I derive: vertical and a.p. GRF are underestimated by all the methods. May an „amplification“ factor give results near to the reference? And under this normalization, which method gives best results?
m.l. GRF due to geometry of camera positions may not be estimated better. I would skip the graphs – you claim only estimation in 2D, the lateral movements are not in the scope of your study, why should you address them?

Why do you in the introduction address just sport sciences (with their fast motions)? The methods are as well of interest for clinical analyses. And in all days‘ clinical life, fast and simple test routines are of high value, e.g. for a triage whether to invest more time and thus money in application of detailled diagnostic methods.

Author Response

We thank the reviewer for their time and comments related to our work. For ease of reference, we have specifically replied to each reviewer comment below, highlighting any undertaken changes in the revised version of the manuscript.

The authors claim to estimate GRF, and not to measure them – under this perspective I read the manuscript. But we have to keep in mind, that estimation is just a measurement with lower demands on what concerns accuracy. And I do not find strainable remarks on the accuracy, nor what concerns the demands ex ante neither for the results ex post the experiments.

We adhered to the following definition of estimation, prediction, and forecasting, which are all terms frequently used in machine learning: “Estimation implies finding the optimal parameter using historical data whereas prediction uses the data to compute the random value of the unseen data. […] Forecasting problems are a subset of prediction problems wherein both use the historical data and talk about the future events. The only difference between forecasting and prediction is the explicit addition of temporal dimension in forecasting.” Since we are building a model, i.e., we are finding the most appropriate parameters to describe a multivariate distribution of historical data, the term “estimation” is used appropriately. We are then testing the model on unseen data, which could be described as a prediction, however, the model building is the focus of this study. The GRF we use as ground-truth for comparison and to build the model has been measured using the gold-standard instrument, a ground-embedded force plate. The accuracy of the model is detailed in Table 2 and Section 4.2.

For readers from outside biomechanics (the journal is Sensors), some basics on the state of the art of measurement of GRF would be senseful.

A basic explanation on the working principle of a force plate has been added to the Introduction: “Ground embedded force plates have been used in biomechanics for decades and are the current gold-standard method to quantify external forces during human movement. Tri-axial force transducers are mounted in each corner of the force plate, that determine the three-dimensional (3D) force. The four single forces are then used to determine the resulting force in vertical, medio-lateral and anterior-posterior direction. Further, the measurement of the four sensors in the corners allows for the calculation of the centre of pressure and free moment of rotation applied to the force plate.” Further information can be found in e.g., Nigg & Herzog, 1994.

From perspective of this state of the art, an effective sampling rate of 8.02 FPS to 16.62 FPS is even too low for analyses of walking, the more for running.

The video cameras used in this study captured interlaced videos with a frame rate of 25 frames per second, which resulted in deinterlaced videos with 50 fields per second. The sampling rate stated here refers to the achieved frame rate when processing videos with different pose estimation algorithms. This is independent of the frame rate of the cameras used, but purely the number of images processed within one second.

This is due to the limitation of sampling frequency of the low budget cameras (I suppose 25 FPS instead of 25 Hz, and fear even those are interlaced),

We used standard video cameras that captured interlaced videos with a frame rate of 25 FPS in this study resulting in deinterlaced videos with 50 fields per second. Most smartphones and cameras used by sports practitioners will not exceed 30 FPS, which results in less information. For the purpose of our study, analysing running speeds of 4.5-5.5 m/s, the frame rate is sufficient.

We changed Hz to frames per second.

and last this is due to and justified by the limited speed of computers used.

Real-time pose estimation was not a goal of this paper. The purpose of referencing the fps were a means for comparison between the pose estimation models. The specs of the computer used are irrelevant for this purpose. Instead, the use of the same computer for all analyses was important, which we achieved. We used a powerful desktop computer with a NVIDIA GeForce RTX 2080, which sits behind the GeForce RTX 2080 Ti as the second fastest desktop GPU in NVIDIA's current Turing line-up. None of the models used the full memory of the GPU.

Not to talk about the limited spatial resolution due to limited number of pixels.

We want to investigate how well the proposed method can be applied to standard videos that can be taken in everyday training. The spatial resolution of 1920x1080~px is common in standard cameras.

I feel the whole story to be very interesting and to be worth to be told to the public soon, scientific progress is driven by studies like these, but I have some minimum demands:

  1. Discussion on measurement errors of the devices, to have quantitative references for the feasible, and possible perspectives for improvements by technology versus methodological limits. For the ATM this is simple, but the problems with the cameras above are complicated by marker displacement on the skin relative to anatomical landmarks.
    Well known in biomechanics, but for the special experiment they have to be quantified. Literature does exist.

Indeed, biomechanical modelling error as a function of soft tissue artifact is a well-known and quantified issue in biomechanics. However, we did not use 3D motion capture techniques in the present study so a discussion surrounding its limitations does not seem warranted in this manuscript.

With respect to the GRF collection instrumentation. The accuracy of the force plate is (https://logemas.com/motion-capture-products/amti-force-plates/):

  • Average COP accuracy of just a fraction of a millimetre (typically less than 0.2)
  • Crosstalk values typically ±0.05% of applied load
  • Measurement accuracy typically ±0.1% of applied load

Again, we did not use any markers in this study. All keypoints are estimated from video data. There is no ground-truth information on joint centres or other anatomical markers for this dataset and this comparison is outside the scope of this study.

  1. 6 is the only figure which gives “time continuous data”. Due to lacking information at the axes and in the legend, I guess the figure shows data from running trials, and the numbers on the horizontal axes represent time in % of cycle. To give the vertical forces in negative values is just unusal (no discussion here on the equivalence of actio to reactio). The information on ISB or Kistler coordinate system is lacking. This has to be improved to refer to existing mental models on GRF.

The figure (Figure 8 in the revised version of the manuscript) has been updated: the vertical ground reaction force is displayed as positive values and the x-axis label has been added. We only investigated running in the neural network application. This was previously only stated in the Introduction and Discussion. The information has now been added to the figure caption, Methods (Section 3.5), and Results (Section 4.2).

And due to the low sampling frequencies, data on walking would be much more interesting than on running what concerns the accurateness – two humps with defined relative heights (the initial spikes and the effects of “wobble masses” in kinematic data are hidden by compliance).

We agree that the analysis of walking gait would be of interest too but is outside the scope of this study. To analyse the validity of the approach for clinical gait analysis, it would be appropriate to include pathological gait (e.g., cerebral palsy, stroke, or osteoarthritis), which will pose a larger challenge to the algorithm. Our dataset was collected on athletes without any movement impairment. We wanted to focus on running as an application in sports, since pose estimation is frequently used in this field. The standard video cameras used in this study provide sufficient video quality for the motion analysed and the sampling frequency of the force plates providing the ground-truth kinetic information is standard in sports biomechanics.

And we need at least one figure for an individual walking and running – if Robin Hood in average hits the centre, his arrows not necessarily hit the disc.

We did not analyse walking. A figure of an individual running trial was added (Figure 7).

No reason is given for the upsampling of motion data to 200 Hz – I guess it is a technical means to synchronize and compare kinematic and force data. But you should explicitly annotate, that by interpolation of kinematic data their quality is not increased.

A statement was added to the Methods (Section 3.4): “The time series were upsampled to enable synchronisation with the force data; this step does not influence the quality of the data.”

From cryptic fig. 6 I derive: vertical and a.p. GRF are underestimated by all the methods. May an “amplification” factor give results near to the reference? And under this normalization, which method gives best results?

Overall, the vertical and anterior-posterior GRF are underestimated (Figure 8) but looking at a single trial (Figure 7) shows that this is not systematic. The use of an amplification factor is therefore not feasible. Moreover, no post-processing of the estimation should be necessary.

m.l. GRF due to geometry of camera positions may not be estimated better. I would skip the graphs – you claim only estimation in 2D, the lateral movements are not in the scope of your study, why should you address them?

The aim of this study was to estimate the 3D GRF from 2D video data. This has been clarified throughout the manuscript. 

Why do you in the introduction address just sport sciences (with their fast motions)? The methods are as well of interest for clinical analyses. And in all day’s clinical life, fast and simple test routines are of high value, e.g., for a triage whether to invest more time and thus money in application of detailed diagnostic methods.

To analyse the validity of the approach for clinical gait analysis, it would be appropriate to include pathological gait (e.g., cerebral palsy, stroke, or osteoarthritis), which will pose a larger challenge to the algorithm. Our dataset was collected on athletes without any movement impairment and therefore does not allow for this analysis. A statement has been added to the Conclusion: “In a clinical context, further validation is necessary with a population showing pathological pose and movement. The transferability of results from this study with a population with healthy pose is most likely limited.”

Round 2

Reviewer 2 Report

line 271: The - the "e" is missing.

Keep on running :-) !